

# Implementing high-throughput insect barcoding in microbiome studies: impact of non-destructive DNA extraction on microbiome reconstruction

Veronika Andriienko[1,2,3], Mateusz Buczek[1], Rudolf Meier[4], Amrita Srivathsan[4], Piotr Łukasik[1] and Michał R. Kolasa[1]

[1] Institute of Environmental Sciences, Faculty of Biology, Jagiellonian University, Krakow, Poland
[2] Doctoral School of Exact and Natural Sciences, Jagiellonian University, Krakow, Poland
[3] Institute of Zoology and Biomedical Sciences, Faculty of Biology, Jagiellonian University, Krakow, Poland
[4] Museum für Naturkunde, Leibniz-Institut für Evolutions- und Biodiversitätsforschung, Berlin, Germany

Corresponding author
Michał R. Kolasa,
michal.r.kolasa@gmail.com

## ABSTRACT

**Background:** Symbiotic relationships with diverse microorganisms are crucial for many aspects of insect biology. However, while our understanding of insect taxonomic diversity and the distribution of insect species in natural communities is limited, we know much less about their microbiota. In the era of rapid biodiversity declines, as researchers increasingly turn towards DNA-based monitoring, developing and broadly implementing approaches for high-throughput and cost-effective characterization of both insect and insect-associated microbial diversity is essential. We need to verify whether approaches such as high-throughput barcoding, a powerful tool for identifying wild insects, would permit subsequent microbiota reconstruction in these specimens.

**Methods:** High-throughput barcoding ("megabarcoding") methods often rely on non-destructive approaches for obtaining template DNA for PCR amplification by leaching DNA out of insect specimens using alkaline buffers such as HotSHOT. This study investigated the impact of HotSHOT on microbial abundance estimates and the reconstructed bacterial community profiles. We addressed this question by comparing quantitative 16S rRNA amplicon sequencing data for HotSHOT-treated or untreated specimens of 16 insect species representing six orders and selected based on the expectation of limited variation among individuals.

**Results:** We find that in 13 species, the treatment significantly reduced microbial abundance estimates, corresponding to an estimated 15-fold decrease in amplifiable 16S rRNA template on average. On the other hand, HotSHOT pre-treatment had a limited effect on microbial community composition. The reconstructed presence of abundant bacteria with known significant effects was not affected. On the other hand, we observed changes in the presence of low-abundance microbes, those close to the reliable detection threshold. Alpha and beta diversity analyses showed compositional differences in only a few species.

**Conclusion:** Our results indicate that HotSHOT pre-treated specimens remain suitable for microbial community composition reconstruction, even if abundance may be hard to estimate. These results indicate that we can cost-effectively combine

barcoding with the study of microbiota across wild insect communities. Thus, the voucher specimens obtained using megabarcoding studies targeted at characterizing insect communities can be used for microbiome characterizations. This can substantially aid in speeding up the accumulation of knowledge on the microbiomes of abundant and hyperdiverse insect species.

# INTRODUCTION

Insects have achieved tremendous evolutionary success, reflected in species diversity estimated in millions, functional diversity, and distribution across almost all terrestrial ecosystems, where they fulfill multiple critically important roles (*Losey & Vaughan, 2006*; *Weisser & Siemann, 2008*). However, their biodiversity is now in steep decline, with habitat degradation, environmental pollution, and climate change identified as some of the key drivers of biomass and diversity losses estimated at 9% per decade, and potentially 40% of all species in the near future (*Sánchez-Bayo & Wyckhuys, 2019*; *Van Klink et al., 2020*). Simultaneously, only a fraction, perhaps one-fifth, of all insect species are known to science (*Stork, 2018*).

The dire need to intensify insect biodiversity characterization and monitoring efforts has not been missed by the scientific community, with the rapid development of approaches such as metabarcoding or high-throughput cost-effective individual barcoding, enabling the characterization of entire insect community samples (*Iwaszkiewicz-Eggebrecht et al., 2023*; *Srivathsan et al., 2024*). On the other hand, the description of approaches for the study of shifting biotic interactions within the monitored multi-species communities has lagged behind. Arguably, the most significant of these associations, yet poorly known outside of a limited set of model species, are those with symbiotic microorganisms: bacteria and fungi that inhabit insect bodies and can dramatically affect their ecology and evolution (*McFall-Ngai et al., 2013*; *Łukasik & Kolasa, 2024*).

The diversity of insects is reflected in at least a comparable diversity of microbial symbionts. Different functional categories of symbionts have diverse and often pivotal effects on the life history traits of their insect hosts, influencing their biology in various ways (*Łukasik & Kolasa, 2024*). Particularly well-known are the microbial roles in the biology and evolution of clades that feed on nutrient-limited diets, including the sap-feeding hemipteran clade Auchenorrhyncha (*Moran, McCutcheon & Nakabachi, 2008*) or blood-feeding insects like bedbugs (*Husnik, 2018*). On the other hand, through their effects on traits such as reproduction and defense against biotic and abiotic factors (*Lemoine, Engl & Kaltenpoth, 2020*), symbionts can strongly influence the biology of insects on much shorter timescales, providing selective advantages to hosts and thus shaping their population dynamics and ecological interactions (*Ferrari & Vavre, 2011*; *Łukasik & Kolasa, 2024*). Some insect groups, including diverse ants (*Sanders et al., 2017*), have not developed symbiotic relationships with specific bacteria but may still harbor

transient microbes derived from food or other environmental sources, yet their overall abundance may be low (*Hammer, Sanders & Fierer, 2019*). Moreover, pathogens can dramatically affect individuals and populations, whether they target insects or are vectored by them (*Dwyer, Dushoff & Yee, 2004*). We know little about the distributions of different functional categories of these microbes within and across multi-species natural communities.

At the same time, insect communities themselves remain poorly characterized in various ecosystems. To address the gaps in our understanding of insect diversity, several initiatives have embarked on high-throughput barcoding ("megabarcoding") (*Chua et al., 2023*) and metabarcoding in the past decade (*Meier et al., 2016*; *Geiger et al., 2016*; *Buchner et al., 2024*). These are improving our understanding of insect community compositions (*Srivathsan et al., 2023*), and efforts have been made to significantly reduce costs and increase throughput in order to conduct such surveys at a large scale effectively (*Meier et al., 2016*; *Srivathsan et al., 2021*). However, to go beyond characterizing insect communities, to addressing questions about insect microbial diversity, distribution, and roles across large multi-species insect collections, whether representing particular clades or community samples, we need robust, cost-effective workflows that simultaneously provide insect identity information with unbiased picture of their microbiota.

Most studies on insect microbiomes examine target species of interest, but attempts to address microbiome-related questions at the level of multi-species community have been rare, with no consensus on the methods (*Kolasa et al., 2019*; *Nakabachi, Inoue & Hirose, 2022*). Some authors attempted surveying microbiota using the whole bulk multi-species samples, the same as those used for insect metabarcoding (*Gibson et al., 2014*). This is, however, limiting given that it dissociates the microbial data from specimen identity. The alternative strategy comprises processing insect specimens individually (*Srivathsan et al., 2023*). Recent megabarcoding approaches aim to obtain reference DNA barcodes for insect specimens using non-destructive DNA extraction approaches that give sufficient template DNA for PCR while preserving morphological features and much of the genomic DNA within the specimen (*Srivathsan et al., 2024*). This is usually achieved by treating the specimen with alkaline buffers to leach the DNA, which can then be used as a template for PCR with broad-spectrum primers, often for the mitochondrial cytochrome oxidase subunit I (COI) gene. The sequenced amplicon products enable the reconstruction of the barcode, used to group the physical specimens into mOTUs (molecular Operational Taxonomic Units) for possible further morphological examinations. However, such barcoded specimens could also be used to obtain associated microbial data.

16S rRNA amplicon sequencing has been a popular technique for characterizing microbial community composition across multiple samples (*Langille et al., 2013*). On the other hand, marker gene analysis may miss some taxa due to amplification biases and provide limited information on microbiome functional capabilities. Metagenomics, in contrast, can provide detailed information on the taxonomic position and function of more abundant symbionts, but its higher cost, computational demands, and analytical challenges limit broad implementation (*Knight et al., 2018*). For such methods, we want to select samples carefully. Then, ideally, when addressing questions about microbiota across

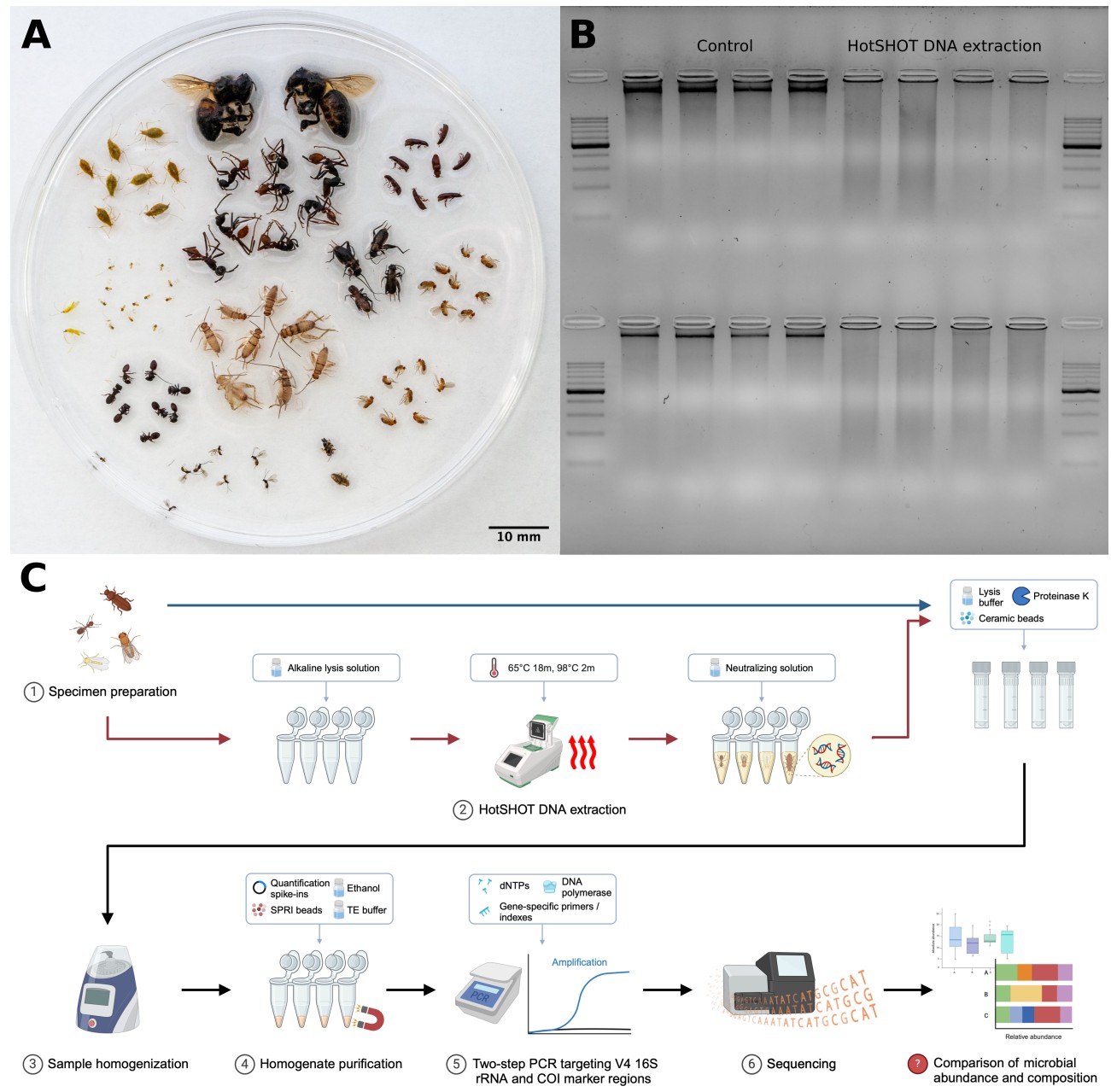

**Figure 1 Insect species used in the study, DNA quality comparison, and study design.** (A) Representatives of the insect species used in the analysis. (B) Gel electrophoresis comparison of DNA extracted from two insect species, *Acyrthosiphon pisum* (first row) and *Aphis fabae* (second row), between HotSHOT-treated and untreated samples. In the control samples, the DNA appears as long fragments, while in the HotSHOT-treated samples, the DNA appears as a smear of different fragment lengths typically shorter than 1 kb, indicative of substantial degradation (ladder used: EURx Perfect™ 100–1,000 bp DNA ladder). (C) Experimental workflow (Created with BioRender.com).

diverse insects (Fig. 1A), we would want to combine methods: first, survey insect diversity broadly and then, based on survey results, select specimens for a thorough characterization using more comprehensive methods. However, a key question is whether the different methods can plausibly be applied to the same specimen one after another. In order to

**Table 1 Insect species used in the study.**

| Order | Family | Species | Origin | Number of specimens (HotSHOT treated/control) |
|---|---|---|---|---|
| Hemiptera | Aphididae | *Acyrthosiphon pisum* | Cultured clone | 7/11 |
| Hemiptera | Aleyrodidae | *Aleyrodes proletella* | Colony from a single plant | 13/10 |
| Hemiptera | Aphididae | *Aphis fabae* | Colony from the single plant | 5/5 |
| Hymenoptera | Apidae | *Apis mellifera* | Social insect colony | 7/2 |
| Hymenoptera | Formicidae | *Cephalotes varians* | Social insect colony | 6/5 |
| Hymenoptera | Braconidae | *Dacnusa sibirica* | Commercial culture | 8/9 |
| Hemiptera | Delphacidae | *Dicranotropis hamata* | Wild-caught from a population | 7/1 |
| Diptera | Drosophilidae | *Drosophila melanogaster* | Laboratory culture (2 lines) | 11/20 |
| Diptera | Drosophilidae | *Drosophila simulans* | Laboratory culture (2 lines) | 26/21 |
| Diptera | Drosophilidae | *Drosophila teissieri* | Laboratory culture (2 lines) | 3/20 |
| Hymenoptera | Formicidae | *Eciton burchellii* | Social insect colony | 7/5 |
| Orthoptera | Gryllidae | *Gryllodes sigillatus* | Commercial culture | 7/10 |
| Orthoptera | Gryllidae | *Gryllus bimaculatus* | Commercial culture | 7/10 |
| Hemiptera | Miridae | *Macrolophus pygmaeus* | Commercial culture | 6/4 |
| Blattodea | Blattidae | *Shelfordella lateralis* | Commercial culture | 7/10 |
| Coleoptera | Tenebrionidae | *Tribolium confusum* | Laboratory culture | 6/10 |

combine insect community characterization with microbiome characterization of the same insects, it is important to assess whether the voucher material generated during megabarcoding can be used for microbiome characterization, as extraction methods may have an effect on DNA quality, impacting the amount and reliability of obtained information (Fig. 1B).

Here, we assessed the impact of the non-destructive DNA extraction method, HotSHOT, used in popular high-throughput barcoding protocols for insect diversity surveys (*Srivathsan et al., 2021*, *2023*; *Vasilita et al., 2024*; *Hu et al., 2024*), on the reconstructed microbial community composition. Specifically, we asked whether microbial abundance estimates and the reconstructed bacterial community profile may change due to HotSHOT pre-treatment.

We did this by comparing quantitative V4 16S rRNA amplicon sequencing data for 286 specimens of 16 species, whether HotSHOT-treated or not (Figs. 1A, 1C). Our findings of the limited effects of HotSHOT/barcoding pre-treatment on insect microbial community profiles open up new avenues for monitoring the diversity and distribution of both insects and their microbial associates.

# MATERIALS AND METHODS

## Specimen collection and preparation

Portions of this text were previously published as part of a preprint (*Andriienko et al., 2024*). For the study, we selected 16 insect species representing six orders (Table 1). Species were selected based on the expectation of relative compositional homogeneity in

microbiota among individuals while acknowledging the potential natural variation caused by independent factors, such as sex, age, or physiological state. They originated from standardized commercial or laboratory cultures, social insect colonies, clusters of hemipteran insects collected from the same plants and likely closely related or clonal, and in just one case (*Dicranotropis hamata*), comprised different and likely unrelated individuals from a single population (Table 1). For three *Drosophila* species, we used distinct lines, one *Wolbachia*-positive and another *Wolbachia*-negative; for simplicity of the narrative, we will include these lines in the count of "species" further on. For all insects, but especially those not originating from commercial or laboratory cultures, we confirmed their identity by comparing barcodes with a customized reference database based on MIDORI (*Leray et al., 2018*). All the specimens of a species were at the same developmental stage but with some variation in body shape, cuticle properties, and size in at least some cases. The specimens were preserved in 95% ethanol and stored at −20 °C. Before the procedure, the size of each individual was measured and categorized into five groups–1 (up to 2 mm), 2 (up to 5 mm), 3 (up to 1 cm), 4 (up to 2 cm), 5 (more than 2 cm). We aimed to use eight–12 individuals per species per treatment, but because of specimen availability, and the loss or damage of some during the barcoding treatment and shipment, it was not possible in all cases.

## Lysis and DNA extraction

Collected insects were divided into two groups. From one group, DNA was extracted at the National University of Singapore using the HotSHOT method (*Truett et al., 2000*). The insects were placed in a 96-well plate filled with 10–15 μl of alkaline lysis solution (25 mM NaOH, 0.2 mM $Na_2EDTA$, pH 12) individually and then incubated in a thermocycler for 20–18 min at 65 °C and 2 min at 98 °C (Fig. 1C). In the case of larger insects, more lysis solution was added, although complete submersion was not necessary. Once heated, the DNA extract was neutralized by adding an equal volume of neutralization buffer (10–15 μl of 40 mM Tris-HCl) (*Truett et al., 2000*; *Srivathsan et al., 2021*). The DNA extract was then immediately used for the barcoding procedure, as described in *Srivathsan et al. (2021)*, and the insects were shipped to Jagiellonian University (JU), Poland, for further processing.

From that point onwards, all insects (those HotSHOT-pretreated and untreated controls) underwent the same processing procedure. For homogenization, the insects were placed in 2 ml tubes filled with 200 μl of a buffer mixture consisting of 195 μl 'Vesterinen' lysis buffer (0.4 M NaCl, 10 mM Tris-HCl, 2 mM EDTA pH 8, 2% SDS) (*Aljanabi & Martinez, 1997*; *Vesterinen et al., 2016*) and 5 μl proteinase K. More solution was added if the sample was not completely submerged. After adding 2.8 and 0.5 mm ceramic beads, the tubes were grounded in the Omni Bead Raptor Elite homogenizer for two 30-s cycles, with the speed set to 5 m/s. Samples were then incubated in a thermal block at 55 °C for 2 h.

Once cooled, 40 μl of homogenate from each tube was transferred to a deep-well plate, where to each, we added a specific number of copies of a quantification spike-in (Table S1) —a linearized plasmid carrying an artificial 16S rRNA target Ec5001 (*Tourlousse et al., 2017*) suspended in 2 μl of TE buffer. The DNA was then purified with 80 μl of SPRI beads using a magnetic stand and washed twice with 80% ethanol. After dilution with 20.5 μl of

TE buffer, 20 µl of the solution was transferred to a new 96-well plate, and the DNA concentration was measured with the Quant-iT PicoGreen kit.

## Library preparation and sequencing

The amplicon libraries were prepared following a custom two-step PCR protocol (*Kolasa et al., 2023*; *Mulio et al., 2024*). The first step involved simultaneous amplification of two marker regions: a V4 region of the 16S rRNA bacterial gene and a portion of an insect mitochondrial cytochrome oxidase I (COI) gene. We used template-specific primer pairs with Illumina adaptor tails: for the V4 region, 515F (GTGYCAGCMGCCGCGTAA) and 806R (GGACTACNVGTWTCTAAT) (*Parada, Needham & Fuhrman, 2016*), and for COI, BF3 (CCHGAYATRGCHTTYCCHCG) and BR2 (TCDGGRTGNCCRAAR AAYCA) (*Elbrecht et al., 2019*). The PCR solution consisted of 5 µl of QIAGEN Multiplex Master Mix, a mix of primers at concentrations 2.5 uM (COI) and 10 uM (16S V4), 2 µl of DNA template, and 1 µl of water (final volume: 10 µl). The program for the first round of PCR included the initial step of denaturation at 95 °C for 15 min, followed by 25–27 cycles of denaturation (30 s, 94 °C), annealing (90 s, 50 °C) and extension (90 s, 72 °C) phases, and the final extension step (10 m, 72 °C). The products were checked on 2.5% agarose gel against positive and negative controls and cleaned with SPRI beads.

During the second indexing PCR, Illumina adapters and unique index pairs were added to the samples. The temperature program for this PCR remained the same, but the number of cycles was reduced to seven. As in previous steps, positive and negative controls (for both PCRs) were included to verify accuracy.

The libraries were pooled approximately equimolarly based on band intensity on agarose gels to ensure a roughly equal representation of each sample in the pool. After the last cleaning step with SPRI beads, the pools were sequenced on an Illumina MiSeq v3 lane (2 × 300 bp reads) at the Institute of Environmental Sciences (JU).

## Bioinformatic processing

The bioinformatics analysis of the data was performed on a Unix cluster using a pipeline developed in the Symbiosis Evolution Research Group, combining custom Python scripts with already established bioinformatics tools, and outlined before (*Kolasa et al., 2023*; *Mulio et al., 2024*).

First, the amplicon data in FASTQ format were split into separate bins based on the primers using a dedicated script, which split the data based on used primer sequences into bins representing marker genes of interest (COI, 16SV4). As all of the specimens used in this study were of known species, we focused only on the 16S rRNA data in further steps in the analysis.

Initially, using PEAR, we assembled quality-filtered forward and reverse reads into contigs (*Zhang et al., 2014*). Next, contigs were de-replicated (*Rognes et al., 2016*) and denoised (*Edgar, 2016*) separately for every library to avoid losing information about rare genotypes that could happen during the denoising of the whole sequence set at once (*Prodan et al., 2020*). The sequences were then screened for chimeras using USEARCH and classified by taxonomy using the SINTAX algorithm and customized SILVA database

(version 138 SSU) (*Quast et al., 2013*). Finally, the sequences were clustered at a 97% identity level using the UPARSE-OTU algorithm implemented in USEARCH. The tables with two levels of classification were produced: ASVs (Amplicon Sequencing Variant) (also known as zOTUs—zero-radius Operational Taxonomic Units) describing genotypic diversity and OTUs (Operational Taxonomic Units)—clustering genotypes based on a similarity threshold.

Bacterial 16S rRNA gene data were screened for putative DNA extraction and PCR reagent contaminants using negative controls (blanks) for each laboratory step as a reference. We first used taxonomy classification information to remove genotypes classified as chloroplasts, mitochondria, Archaea, or chimeras. Next, we calculated relative abundances and used ratios of each genotype presented in blank and experimental libraries to accurately assign genotypes as putative actual insect-associated microbes or PCR or extraction contaminants (Table S2).

Additionally, reads identified as quantitative spike-ins were used to reconstruct bacterial absolute abundances in the processed insects. Specifically, the symbiont-to-extraction spike-in ratio, multiplied by the number of extraction spike-in copies and the proportion of the homogenate, allowed us to estimate amplifiable bacterial 16S rRNA copy numbers in the homogenized specimens.

Finally, manual analysis was conducted to remove controls, samples with incorrect indexes or zero abundance of bacteria and create the dataset used in the statistical analysis.

## Statistical analysis and visualization

Statistical analysis was performed using the software RStudio v.2023.03.1+446 (*R Core Team, 2023*) and QIIME2 v.2023.2 (*Bolyen et al., 2019*). Inkscape 1.2.2 (*Inkscape Project, 2022*) was used to modify generated plots and visualizations.

One-way ANOVA with random effects on insect species was used for the absolute abundance comparison between groups treated with HotSHOT and those that were only homogenized. The base and 'nlme' (*Pinheiro, Bates & R Core Team, 2023*) packages were utilized for this step. The analysis was visualized using the packages 'ggplot2' (*Wickham, 2016*), 'dplyr' (*Wickham et al., 2023*), 'RColorBrewer' (*Neuwirth, 2022*) and 'phyloseq' (*McMurdie & Holmes, 2013*).

A biodiversity assessment was performed in QIIME2 on the Unix cluster. Firstly, the decontaminated OTU table, bacteria taxonomy, and file with bacterial OTU sequences in fasta format were imported into QIIME2 as artifacts to allow further transformation and tracking of the origin of the output files of the analysis (*McDonald et al., 2012*). Next, the obtained feature table was filtered separately for each insect species so the differences would not disturb the analysis.

As diversity metrics are sensitive to different sampling depths in the groups, for the biodiversity index comparison we standardized the samples by rarefying them (*Weiss et al., 2017*). Rarefaction level was chosen individually based on the generated summary for these tables, balancing between retaining the highest amount of samples and the percentage of the features left for analysis. Note that all other analyses (absolute and relative abundance

comparisons) were conducted using the original, non-rarefied dataset to avoid additional stochastic biases.

The alpha- and beta-diversity indexes, calculated using the q2-diversity plugin, were used to compare the microbiome composition between the methods for each species. Alpha diversity refers to the diversity within the samples (*Whittaker, 1972*), and *Shannon*'s *(1948)* entropy, which combines richness and evenness evaluation in a single metric and provides a comprehensive assessment of diversity, was chosen for its assessment. The calculated indexes were compared between the groups using the Kruskal-Wallis statistical test (*Kruskal & Wallis, 1952*).

The Bray-Curtis dissimilarity (*Sørensen, 1948*) index was chosen for beta-diversity analysis. This index calculates the dissimilarity between communities based on the presence and abundance of different features. It considers both the presence/absence and the relative abundances of features, providing a quantitative measure of compositional differences between samples.

To visualize the results and explore the beta-diversity patterns, the principle coordinate analysis (PCoA) (*Halko et al., 2011*; *Legendre & Legendre, 2012*) was performed using the Emperor tool (*Vázquez-Baeza et al., 2013*). Moreover, the PERMANOVA and PERMDISP tests were conducted to analyze the statistical trends (*Anderson, 2001*).

## RESULTS

We obtained reliable data for a total of 286 biological samples. We analyzed and interpreted them using an additional 16 negative control samples of different types, representing each of the laboratory steps and batches: six for DNA extraction, seven for the first PCR, and three for the second PCR. Each sample yielded a minimum of 55 16S rRNA reads, ranging from 55 to 93,394 reads, and a mean of 24,100 reads (Table S1).

### The effects of treatment on the microbiome absolute abundance

The quantification spike-in was present in every amplicon library. In experimental samples, the ratio of bacterial reads not classified as contamination to Ec5001 (extraction plasmid) reads ranged from 0.0114 (in *Dacnusa sibirica*) to 16,187.5 (in *Drosophila simulans*). With 10,000 copies of artificial amplification target Ec5001 added to 20% of insect homogenate (in most cases–see Table S1), and assuming no amplification bias among different types of templates, this translates to between 697 and 2,062,562,500 copies of bacterial 16S rRNA per insect. When comparing estimated microbial abundances across species and treatments (two-way ANOVA using log-transformed data), we found significant differences between treatments ($F = 159.86$, $p < 0.0001$). However, the effect of HotSHOT treatment varied among the species (Fig. 2).

We tested the treatment effect in each of the 16 species. The analysis using the Kruskal-Wallis test showed that bacterial abundance was significantly lower after HotSHOT in 13 out of 19 species (Table S3). We also checked if the treatment effect was connected with the insects' size, and test results indicate that it was similarly significant for all size categories that we distinguished (Table S3). On average, the estimated bacterial absolute abundance decreased 15-fold following HotSHOT. We conclude that these values

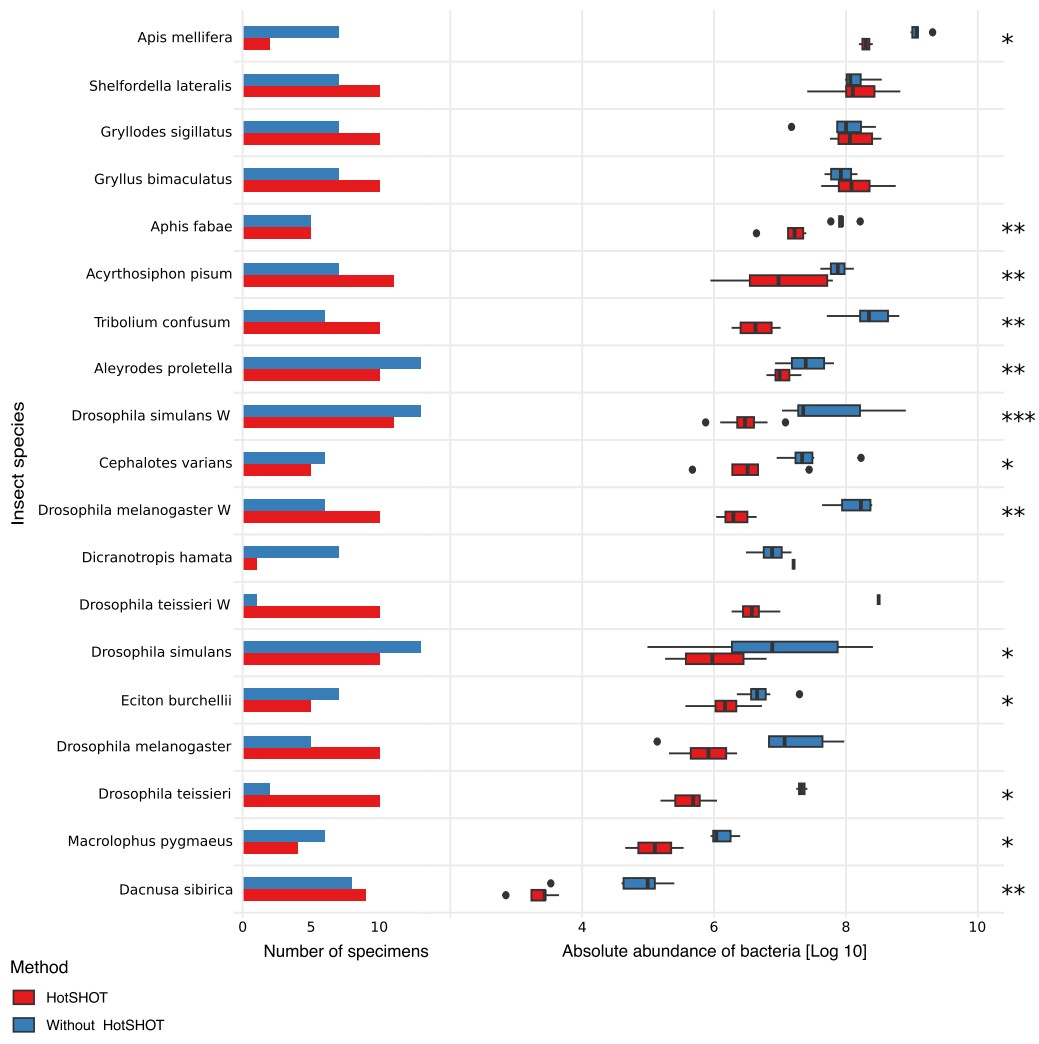

**Figure 2 Comparison of the absolute abundance of bacteria between specimens treated with HotSHOT and control specimens among insect species.** Each bar represents the number of specimens used in the control (blue) and HotSHOT (red) treatments. Box plots represent the absolute abundance calculated for each species and treatment. Asterisks represent the significance levels: * <0.05, ** <0.01, *** <0.001.                                        

represent the actual decrease in the concentration of amplifiable gene targets during incubation in the hot alkaline buffer.

## Microbiome diversity analysis

We obtained biologically realistic data for all species (Fig. 3, Tables S4, S5). As expected, we observed *Acetobacter* and *Lactobacillus* as the dominant symbionts of cultured *Drosophila* species (*Wong, Ng & Douglas, 2011*), *Buchnera* as the dominant symbiont in two aphids (*Douglas, 2003*), and well-known members of gut microbiota of honeybees (*Engel & Moran, 2013*) and *Cephalotes* (*Hu et al., 2018*) and *Eciton* ants (*Mendoza-Guido et al., 2023*). Facultative endosymbiont *Wolbachia* was present in five species and often dominated their microbial communities, whereas *Cardinium* dominated the microbiota of *D. hamata* planthopper. In cultured *Gryllodes sigillatus* and *Gryllus bimaculatus*, we found

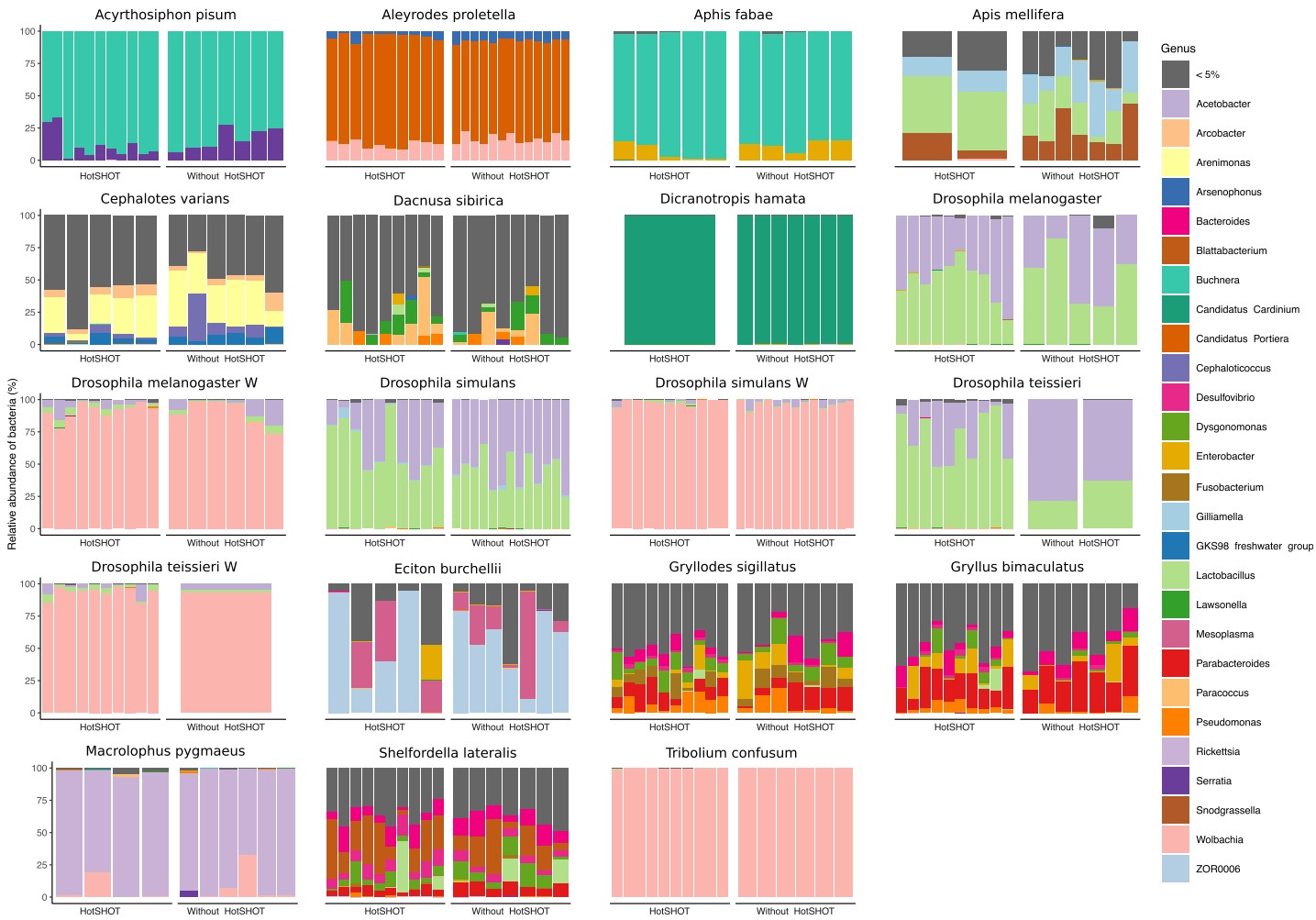

**Figure 3 Relative abundances of bacterial genera in 19 lines of 16 insect species.** Bars represent individual insects, whether HotSHOT-treated or control, of that line. Colors represent microbial genera above 5%, and gray represents the summed abundance of all other genera combined.

relatively complex microbiota comprising relatively widespread bacterial genera as well as some specialists (*Blattabacterium* in *S. lateralis*), at variable abundances. In the species with the least abundant microbiota, parasitic wasp *D. sibirica*, microbiota composition was highly variable.

In three out of 19 lines/species (*Aleyrodes proletella*, *Gryllodes sigillatus*, and *Tribolium confusum*), we observed differences between treatments in richness and evenness in the distribution of features, expressed through significant changes in Shannons' entropy values (Table S6). Beta diversity analysis, conducted using the Bray-Curtis index, revealed significant differences between treatments in microbial community composition in four other species (*Aleyrodes proletella*, *Cephalotes varians*, *Dacnusa sibirica*, and *Drosophila teissieri*). These differences in the presence and relative abundance of features can be observed on PCoA plots for these species (Fig. S1), where the first axis explains from 15.47% to 86.16% of diversity.

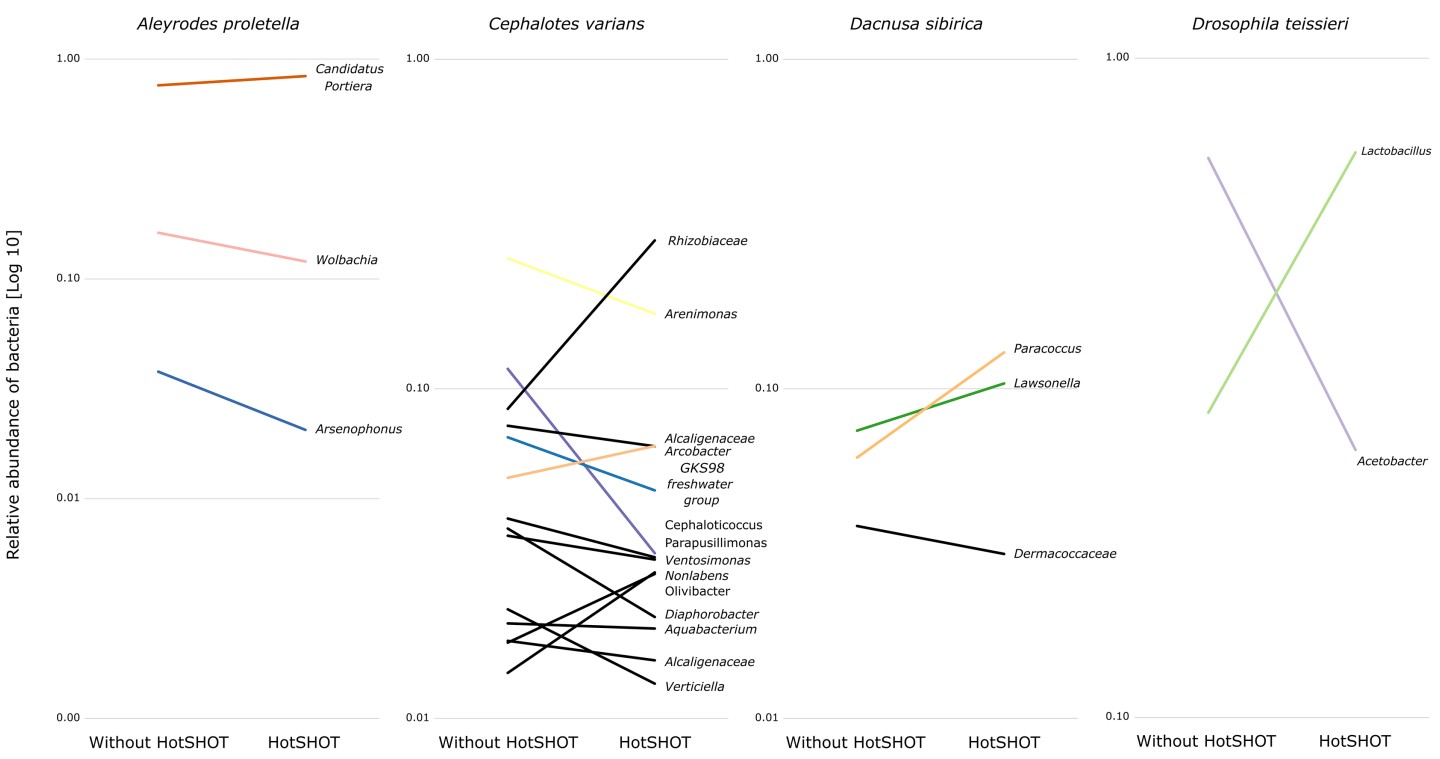

**Figure 4  Changes in the relative representation of microbial genera above 1% between treatments.**

The comparison of average relative abundances of dominant bacterial clades (Fig. 4) shows that the bacterial taxa mostly overlap between the different methods, despite fluctuations in the proportions of some bacterial taxa.

Overall, the microbial communities were dominated by members previously described in particular insect species and, in many cases, by commensals characteristic of the particular species. In general, dominant bacteria were present in all individuals of a species, although their relative abundance varied, reflecting the anticipated natural variation among specimens. Significant differences in composition were not systematic and were related to changes in rarely present bacteria rather than changes in major symbionts. Thus, no overall difference in composition between pre-barcoded and control samples was observed.

## DISCUSSION

The DNA extraction that involves specimen incubation in the hot alkaline buffer, called HotSHOT, affects the DNA integrity and, consequently, the microbial DNA yield within the treated specimens. This results in decreased absolute abundance estimates of 16S rRNA copies in processed specimens, which is presumably a direct consequence of DNA degradation caused by the HotSHOT treatment. Nevertheless, the reconstructed microbial community composition, especially the reconstructed presence of abundant microbial clades, which are likely to play the most significant roles in insect biology, was not substantially affected. Thus, even though the barcoding process disrupts our ability to

estimate the amounts of bacteria colonizing insect bodies, it does not seem to substantially bias conclusions about the identity of these bacteria. However, it has to be kept in mind that pre-treatments such as HotSHOT might decrease the ability to detect less abundant bacteria, such as pathogens vectored by insects. Hence, the combination of HotSHOT pre-treatment and microbiome metabarcoding may not be suitable for all types of microbiome-focused studies.

## Effects on microbial abundance

Insects and other organisms differ dramatically in the abundance of microorganisms they host, and these differences often correlate with the microbes' importance in insect biology (*Hammer, Sanders & Fierer, 2019*). Assuming all else equal, the greater number of microbial cells should translate to their stronger effect, driven by higher nutritional demands and greater amounts of biologically active compounds produced. However, the effects may not be linear due to the bacterial ability to detect their abundance and alter biological activity in response (*quorum sensing*) (*Miller & Bassler, 2001*). For example, some microbial pathogens may delay the production of virulence factors until they sense abundance sufficient for overwhelming host defenses (*Munoz et al., 2020*). On the other hand, in *Sodalis praecaptivus*—a versatile opportunist that seems representative of the ancestral state of many heritable symbionts, *quorum sensing* attenuates virulence, facilitating a long-lasting and benign association (*Enomoto et al., 2017*). From a more technical perspective, microbiome abundance also determines the precision of its reconstruction. We know that low-bacterial-abundance samples are much more prone to contamination from reagents and other sources (*Salter et al., 2014*), likely leading to erroneous conclusions. While deeper sequencing would increase the representation and likelihood of detection of any low-abundance microbes, it would also magnify noise—and in our experience, it can be very hard to separate the two. We can be much more confident about the presence of abundant microbes.

For these reasons, it is important to estimate the absolute abundance of the microbiome rather than just report the relative abundance of microbial clades. Unfortunately, with some notable exceptions (*Hammer et al., 2017*; *Sanders et al., 2017*; *Ravenscraft et al., 2019*; *Surmacz et al., 2024*), it is rarely done in experimental studies. We argue that the available tools for abundance estimation, including quantitative PCRs and spike-ins (*Props et al., 2017*; *Tourlousse et al., 2017*; *Jian et al., 2020*; *Harrison et al., 2021*), should become a part of standard microbiome analyses. Although amplicon-based sequencing of bacterial marker gene fragments might be burdened with various biases (*Gloor et al., 2017*), careful experimental design and the awareness of multiple known caveats (*Knight et al., 2018*) can effectively limit such biases providing a clear picture of insect symbiosis (*Kolasa et al., 2023*; *Mulio et al., 2024*). At the same time, we must be aware that aggressive treatments, such as an incubation in an alkaline buffer at high temperatures, may lead to the degradation of the substantial share of the available DNA and disrupt or complicate abundance reconstruction. Changes to the HotSHOT protocol, such as reduced incubation time, will likely limit DNA degradation while yielding sufficient amounts for barcoding analyses. Indeed, we have recently reduced the routine incubation time to 5 minutes

without compromising barcoding results (V Andriienko & A Michalik, 2024, unpublished data).

## Effects on microbial composition

Organisms differ in which microorganisms they host, and different microbial functional categories, clades, and even strains have very different effects on host biology and evolution (*Bourtzis & Miller, 2003*).

Our study used insect species hosting a broad range of microbial functional categories and taxonomic clades, reflecting the broad range of host ecological niches. Our results are congruent with *a priori* expectations about symbionts present in these species, and HotSHOT treatment has not affected the detection of their dominant microbes. We conclude that our ability to detect abundant bacteria, including nutritional and facultative endosymbionts and specialized gut bacteria known to have major effects on the host ecology, is not being altered by HotSHOT pre-treatment. Although amplicon-based sequencing is not free of errors and can provide a skewed picture of microbiome composition due to factors such as primer bias (*Gloor et al., 2017*), we have found that our amplicon data is congruent with metagenomic data obtained for the same individuals (*Buczek et al., 2024*). However, the ability to detect less abundant but significant microbes following HotSHOT may be reduced. For example, we have recently reported that the agriculturally significant plant pathogen *Phytoplasma* is often represented by only a few reads in 16S rRNA amplicon datasets for individual vector leafhoppers (*Mulio et al., 2024*); the infection signal may be even harder to detect following HotSHOT. Given the known issues with reagents- and cross-contamination, the reliability of detection of such microbes may be limited regardless, even when appropriate controls are used. 16S rRNA amplicon sequencing using broad-spectrum primers may not be the most appropriate tool for surveying *Phytoplasma* infections in wild-caught leafhoppers to begin with! Thus, while HotSHOT treatment in combination with microbiome screening proves to be a valuable tool, it is essential to acknowledge its limitations and the potential biases of amplicon-based sequencing.

Statistical analyses indicate that in a few cases, the HotSHOT treatment introduces bias in the relative abundance of some bacteria taxa. We interpret those results as a likely outcome of the reduction of symbiont DNA amounts caused by the HotSHOT treatment, with reagent contamination thus becoming more pronounced. However, it is on a relatively low level and does not change the overall composition of the insect microbiome. Nevertheless, this fact should not be trivialized, and appropriate negative controls should be implemented and processed to filter out any introduced bacterial signal effectively.

Considering this, the HotSHOT treatment may confound some of the conventional approaches to microbiome analysis, including altering zOTU or OTU counts, rarefaction, or diversity index comparisons. While these approaches may be considered the "gold standard" nowadays, we argue that researchers must carefully consider the biological value and relevance of the information they provide for their system, taking into consideration reagent contamination and other challenges. This is particularly relevant for small organisms such as insects, with often low overall bacterial abundance or dominated by one

of few abundant microbes and with the remainder at low abundance. While this should be decided on a case-by-case basis, we think that analyses focused on abundant insect symbionts—unaffected by the HotSHOT treatment—provide much more reliable and biologically relevant information.

### Broader context

The methods for biodiversity discovery have been shifting, with approaches such as ultra-throughput individual barcoding, or alternatively, metabarcoding, rapidly gaining in popularity. International projects such as BOLD are already processing millions of insect specimens to discover global insect diversity. Thus, we have a growing amount of individuals processed with non-destructive DNA extraction techniques. There is an assumption that these insects remain suitable for further morphological or DNA-based characterization, as their integrity remains preserved. This could prove to be a significant advantage for the field of entomology, given that species definitions rely considerably on an integrative taxonomic approach that combines molecular data with detailed morphological and ecological information (*Dayrat, 2005*; *Pante, Schoelinck & Puillandre, 2015*). However, there are limited examples of successful usage of such pre-processed specimens for addressing further questions.

Our work demonstrates clearly that the associations with some of the most important players in insect biology–symbiotic microorganisms–can be reconstructed reliably, albeit with some caveats, from such pre-processed material. This opens up exciting avenues for microbiota study across large numbers of pre-barcoded wild-caught specimens and cost-effective reconstruction of broad microbiome-related patterns. With bacterial symbionts increasingly regarded as important players in insect biology and a potential source of rapid insect adaptation to changing environments, we advocate that microbiome screening should become a standard procedure in insect biodiversity studies (*Łukasik & Kolasa, 2024*). Hence, rather than attempting labor- and cost-intensive additional sampling for fresh material, relying on microbiome typing on pre-barcoded material could be an extremely frugal way of conducting science. Simultaneously, our approach confirms that the material will likely be suitable for other DNA-based approaches aiming to unravel mechanisms that shape insect diversity and biology.

## CONCLUSIONS

Biodiversity and microbiome researchers have multiple tools that vary in their information output, per-sample cost, and plausible throughput. When addressing biological questions, it is essential to balance these criteria. However, our results suggest that we do not need to limit ourselves to just one tool. On the contrary, by serially applying different methods to the same specimens, we can combine the breadth of extremely cost-effective approaches, such as barcoding, with much deeper insights that could be obtained from multi-gene amplicon sequencing and, likely, also genomics tools. Combining these different tools could become a new paradigm for biodiversity studies in the turbulent era of the Anthropocene.

## ACKNOWLEDGEMENTS

We thank Brandon Cooper and Jacob Russell for providing some of the specimens that were used in this study.

### Funding

The project was supported by the Polish National Agency for Academic Exchange grant PPN/PPO/2018/1/00015, the Polish National Science Centre grants 2018/31/B/NZ8/01158 and 2021/43/B/NZ8/03376, the National Institute of General Medical Sciences of the NIH award number R35GM124701 to Brandon S. Cooper (insect culture), and Jagiellonian University POB BioS minigrant (ID: B.1.11.2020.101). The APC of this article was supported by the subsidy for scientific activities of the Jagiellonian University No. N18/DBS/000023. The funders had no role in study design, data collection and analysis, decision to publish, or preparation of the manuscript.

### Grant Disclosures

The following grant information was disclosed by the authors:
Polish National Agency for Academic Exchange: PPN/PPO/2018/1/00015.
Polish National Science Centre: 2018/31/B/NZ8/01158 and 2021/43/B/NZ8/03376.
National Institute of General Medical Sciences of the NIH: R35GM124701.
Jagiellonian University POB BioS minigrant: B.1.11.2020.101.
Jagiellonian University: N18/DBS/000023.

### Competing Interests

The authors declare that they have no competing interests.

### Author Contributions

- Veronika Andriienko performed the experiments, analyzed the data, prepared figures and/or tables, authored or reviewed drafts of the article, and approved the final draft.
- Mateusz Buczek performed the experiments, authored or reviewed drafts of the article, and approved the final draft.
- Rudolf Meier conceived and designed the experiments, authored or reviewed drafts of the article, and approved the final draft.
- Amrita Srivathsan performed the experiments, authored or reviewed drafts of the article, and approved the final draft.
- Piotr Łukasik conceived and designed the experiments, analyzed the data, prepared figures and/or tables, authored or reviewed drafts of the article, and approved the final draft.
- Michał R. Kolasa conceived and designed the experiments, analyzed the data, prepared figures and/or tables, authored or reviewed drafts of the article, and approved the final draft.

## DNA Deposition

The following information was supplied regarding the deposition of DNA sequences:

All data underlying this study are available at the Sequence Read Archive (SRA) of the National Center for Biotechnology Information (NCBI): PRJNA1102268.

## Data Availability

The pipeline documentation is available at GitHub:

- https://github.com/Symbiosis-JU/Bioinformatic-pipelines/blob/main/Example_analysis.md.

The versions of the scripts used in the current analysis are available at GitHub and Zenodo:

- https://github.com/NAndriienko/HotSHOT-Microbiome-Project.

- Andriienko, V., & Kolasa, M. (2024). Implementing high-throughput insect barcoding in microbiome studies: impact of non-destructive DNA extraction on microbiome reconstruction - bioinformatic pipeline. Zenodo. https://doi.org/10.5281/zenodo.13329523.

## Supplemental Information

Supplemental information for this article can be found online at http://dx.doi.org/10.7717/peerj.18025#supplemental-information.

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
