# Peer review of "Implementing high-throughput insect barcoding in microbiome studies: impact of non-destructive DNA extraction on microbiome reconstruction"

_PeerJ, doi:10.7717/peerj.18025_

## Round 0.1 · original submission · Major Revisions

Please address the concerns raised by each reviewer

Reviewer 1 ·

Basic reporting

The work “Implementing high-throughput insect barcoding in microbiome studies: impact of non-destructive DNA extraction on microbiome reconstruction” (#100348) focuses on the possibility of using the same samples for both insect barcoding and bacterial metabarcoding.

The aim and scope of the work are very valuable, especially in the context of a better understanding of quantitative metabarcoding analysis, and whether some techniques may impact the results.

The results obtained are important, but I think the authors could go more in depth in some of the difference recorded between treatments. In general, the manuscript is well written and clear, except for a full paragraph of the discussion that is more a reflection of literature review since it is not mentioning any results from the paper.

Some additional reference should be added to the discussion, as mentioned below.

Experimental design

The research is within the aims and scope of the journal and the authors tackle an important topic.

The experimental design appears to have a very clear flaw, which does not seem to be addressed enough:
- Why did the authors use different numbers of individuals for the two treatments for 18/19 of the species? In the methods they explained that “Species were selected based on the expectation of relative homogeneity in microbiota composition among individuals.”, and if this was true, then it would make sense to then assume that a decrease in bacterial abundance between treatments was indeed due to the treatment itself. However, Figure 2 and Figure 4 clearly show there can be a very high variability in bacterial absolute abundance across the individuals of many species, even in the control. Then why didn’t the authors try to limit this additional bias by using the same number of insects for both the treatment and the control? Also, how do the authors explain such a massive variation in absolute abundance when they expected these individuals to have close to no variation?
At lines 196-199 the authors mentioned they have rarefied their dataset, are Figure 2 and Figure 4 showing the results post-rarefaction?

Validity of the findings

In order to better understand the true validity of these findings, I have a few questions that should be addressed:

- What is the effect on the chances to record bacterial rare species? The authors have found that the hot-shot method impacts our ability to estimate the amounts of bacteria colonising insect bodies by decreasing the number of reads for the abundant bacterial taxa. However, the authors suggest that this process does not seem to “substantially bias conclusions about the identity of these bacteria” (Line 281). I suspect this is the case for those bacterial taxa that showed 1,000,000 of reads in the control and ended up with 100,000 reads in the hot-shot sample.. but what happened to the bacterial taxa that had only 10,000 reads to start with? What to the taxa with 2,000 reads? What was the limit threshold under which bacteria could not be recorded anymore? I think this is an extremely important aspect that needs to be discussed. I strongly disagree with the discussion at lines 323-327. To say that only the abundant microbes are important because “low-abundance microbes, even if reliably detected, are also less likely to form biologically relevant associations significant to the host” is not true and it is deflecting the issue. For example, what if low-level pathogens are present in the insect? What’s the threshold under which we would not be able to record them if using hot-shot? I think the authors have a good dataset in hand, and they just have to cross-check what bacterial OTUs have disappeared between the control and the hot-shot and determine what was the original read-count of these OTUs. This will provide a reliable threshold.

Lines 282-312 “Effects on Microbial abundance”.
This whole paragraph does not discuss the results of the paper. These papers can be mentioned in the introduction, but not in the discussion. Furthermore, please consider the following comments. While it is true that the absolute abundance would be a great way to assess bacterial diversity, the authors are limited by their choice to use metabarcoding. This technique does not allow to confidently measures absolute abundance, due to the compositional nature of the approach, and the multitude of biases introduced at every step of the analysis.

Line 323-327: As mentioned above, this is not true. There are many instances where the “abundance” reported by the number of reads is not a real reflection of reality. The issue of bias in metabarcoding results has been widely studied and the fact that abundant reads = abundance bacteria has been disproved multiple times. For example, primer bias may impact what bacterial OTUs are the most recorded. There are studies showing that the number of reads for the primary symbiont (most important bacteria that allows the insect to survive) of hemipteran insects are constantly low, and it is probably due to the primer bias.
I suggest the article by Gloor and colleagues (Gloor et al. 2017, Front. Microbiol., Sec. Systems Microbiology, Volume 8 - 2017 | https://doi.org/10.3389/fmicb.2017.02224 ) that discusses this in depth.

Lines 328-329: “Statistical analyses indicate that in a few cases, the HotSHOT treatment elevates the abundance of some bacteria taxa”.
This is another instance highlighting the issue the authors are facing. Obviously, it is impossible that a hot-shot treatment “elevates the abundance of some bacteria”. What the hot-shot treatment is doing is biasing the RELATIVE ABUNDANCE of those certain bacterial OTUs. This treatment is impacting what reads can/cannot be recorded. The sentence needs rephrasing and the discussion needs to discuss more in depth the metabarcoding bias, the relative abundance of reads and the compositional nature of metabarcoding.

Additional comments

Minor comments:

Line 141: the authors should specify the type and number of negative controls. At line 216 they have mentioned “16 negative control samples of different types” but these have not been clearly listed anywhere.

Line 183: Rephrase this sentence “Statistical analysis was performed using RStudio version 2023.03.1+446 (R Core Team, 2023) and QIIME2 2023.2 (Bolyen et al., 2019) software”
this way: “Statistical analysis was performed using the software RStudio v.2023.03.1+446 (R Core Team, 2023) and QIIME2 v.2023.2 (Bolyen et al., 2019)”

Line 189: Rephrase this sentence “The analysis was visualized with the usage of ‘ggplot2’ (Wickham, 2016), dplyr (Wickham et al., 190 2023), RColorBrewer’ (Neuwirth, 2022) and phyloseq (McMurdie & Holmes, 2013) packages.
This way: “The analysis was visualized using the packages ‘ggplot2’ (Wickham, 2016), ‘dplyr’ (Wickham et al., 2023), ‘RColorBrewer’ (Neuwirth, 2022) and ‘phyloseq’ (McMurdie & Holmes, 2013).

Line 250: What “more formal analyses” are the authors referring to?

Reviewer 2 ·

Basic reporting

Implementing high-throughput insect barcoding in microbiome studies: impact of non-destructive DNA extraction on microbiome reconstruction (#100348)

Dear Editor,

This manuscript investigated whether bacterial community of a small set of selected insect species can be study using an approach in which the specimens remain intact versus a non-destructive approach. The methodology is suitable for this purpose, but I found the manuscript a bit confusing because the authors imply in the text that their approach can be used to simultaneously identify insect species and they bacterial community. In the Material and Methods, the authors mentioned that they built their library including COI, but they do not mention anything related to that in the Results. I cannot understand how they determined the insect species taxonomic identity of each specimen used in the statistical test to compare bacterial abundance? Was it by checking the unique index pairs of the samples identified as bacteria and associating it with the corresponding insect species? Or did thy blast their COI sequences against a database to obtain taxonomic information? This is the main drawback of the study. Bellow are further comments that I think might improve the manuscript.

I could not understand your gel image in Figure 1. What is the difference between the two rows of wells? What is the content of individual wells? It appears to me that there is no DNA at all in the “HotSHOT DNA extraction” wells. You do not properly explain this image in the text and you mention Figure 1B in the Introduction section before Figure 1A. You should provide more information in this figure and cite Figure 1A before Figure 1B.
I can understand Figure 3, but the image quality is poor. This is also valid for Figure 2, although it is much better that Figure 3. It is difficult to read the taxonomic names. These bar plots can be easily done using ggplot2 and perhaps you should try to produce a new figure in vector format. This will greatly increase figure’s quality. Figure 4 is also of poor quality.
Line 57. You forgot to delete this: “(Fig. 1B)”
Line 224. This is difficult to understand. Perhaps you should use commas and dots to separate the decimals
325-327 – Are you capable of backing up this assumption with a reference?
Discussion section: I agree that molecular methods are helping to identify patterns of insect diversity worldwide, but please note that the results you get from the sequencing are not real insects. The study of biodiversity of insects is different from that of bacteria. Insects are studied in details and the definition of species relies on an integrative taxonomic approach. BOLD is an amazing initiative that has allowing biodiversity assessment through molecular methods, but you should have in mind that their database is far from including all taxonomic described species, even after more than 20 years. This should be addressed in you discussion section, perhaps in the “Broader context” subsection.

Experimental design

See basic report.

Validity of the findings

See basic report.

Additional comments

See basic report.

Reviewer 3 ·

Basic reporting

Paper fulfill all requirements for reporting.

Experimental design

Experimantal design is well chosen, particularly the use of samples where low variation between individuals can be expected.

Validity of the findings

The analyses performed and the way they are interpreted convinced me of the validity of the findings.

Additional comments

This is a solid paper on an important and timely topic. Megabarcoding is a very promising technique for large scale surveys. To be the most efficient and as cheap as possible, it relies on rapid hotshot extraction. It is thus important to explore how usable are the remnants of the samples after hotshot extraction for additional applications, including the microbiome sequencing explored in this paper. The result of the tests is a yes with some caveats in microbe abundance quantification, which means this paper will likely have an impact on how insect microbiome surveys are performed.

The paper is generally well-written. The introduction is clear and includes the main aspects of the background I would expect. For methods, the logic of choosing mostly samples where low intraspecific variation is expected is a neat approach. The molecular methods are solid, use of spike-in allows quantification, multiple types of negative controls sequenced, etc. as appropriate.

The same goes for results, figures and discussion. The paper feels like it already went through some round of peer review, I don’t find any major issue, and extremely minimal number of minor points.

Minor points:
L92: this is first mention of that species so please include genus name in full
L297-300: not just contamination, but also simple sampling effect influences characterization of communities
L314: Better phrasing could be: “Organisms host different microrganisms with various function,…”
L323-327: Could deeper sequencing of hotshot treated samples help, or are you convinced that it is purely signal/noise problem?

---

## Round 0.2 · accepted · Accept

The authors have addressed all reviewers comments and the paper is ready for publication